# Synthesis, In Silico, and In Vitro Evaluation of Anti-Leishmanial Activity of Oxadiazoles and Indolizine Containing Compounds Flagged against Anti-Targets

**DOI:** 10.3390/molecules24071282

**Published:** 2019-04-02

**Authors:** Strahinja Stevanovic, Milan Sencanski, Mathieu Danel, Christophe Menendez, Roumaissa Belguedj, Abdelmalek Bouraiou, Katarina Nikolic, Sandrine Cojean, Philippe M. Loiseau, Sanja Glisic, Michel Baltas, Alfonso T. García-Sosa

**Affiliations:** 1Laboratory for Bioinformatics and Computational Chemistry, Institute of Nuclear Sciences VINCA, University of Belgrade, P.O. Box 522, 11001 Belgrade, Serbia; strahinja.stevanovic@protonmail.com (S.S.); sencanski@vin.bg.ac.rs (M.S.); 2ITAV, Université de Toulouse, CNRS, 31062 Toulouse, France; mathieu.danel@itav.fr; 3Department of Chemistry, Université de Toulouse, UPS, CNRS UMR 5068, LSPCMIB, 118 Route de Narbonne, 31062 Toulouse, France; menendez@chimie.ups-tlse.fr (C.M.); racha_belg@hotmail.fr (R.B.); 4CNRS, Laboratoire de Synthèse et Physico-Chimie de Molécules d’Intérêt Biologique, LSPCMIB, UMR-5068, 118 Route de Narbonne, 31062 Toulouse, France; 5Unité de Recherche de Chimie de l’Environnement et Moléculaire Structurale, Université Frères Mentouri, Route de Ain El Bey, 25000 Constantine, Algeria; bouraiou.abdelmalek@yahoo.fr; 6Department of Pharmaceutical Chemistry, Faculty of Pharmacy, University of Belgrade, Vojvode Stepe 450, 11000 Belgrade, Serbia; knikolic@pharmacy.bg.ac.rs; 7Antiparasitic Chemotherapy, UMR 8076 CNRS BioCIS, Faculty of Pharmacy Université Paris-Sud, Rue Jean-Baptiste Clément, F 92290 Chatenay-Malabry, France; sandrine.cojean@u-psud.fr; 8Institute of Chemistry, University of Tartu, Ravila 14a, 50411 Tartu, Estonia

**Keywords:** Leishmania, arginase, in silico, anti-target, in vitro, anti-leishmanial inhibitors, anti-target

## Abstract

Due to the lack of approved vaccines against human leishmaniasis and the limitations of the current chemotherapy inducing side effects and drug resistance, development of new, effective chemotherapeutic agents is essential. This study describes the synthesis of a series of novel oxadiazoles and indolizine-containing compounds. The compounds were screened in silico using an EIIP/AQVN filter followed by ligand-based virtual screening and molecular docking to parasite arginase. Top hits were further screened versus human arginase and finally against an anti-target battery to tag their possible interactions with proteins essential for the metabolism and clearance of many substances. Eight candidate compounds were selected for further experimental testing. The results show measurable in vitro anti-leishmanial activity for three compounds. One compound with an IC_50_ value of 2.18 µM on *Leishmania donovani* intramacrophage amastigotes is clearly better positioned than the others as an interesting molecular template for further development of new anti-leishmanial agents.

## 1. Introduction

The leishmaniases are a group of diseases caused by the trypanosomatid protozoan parasite of the genus *Leishmania* through the bite of infected phlebotomine sandflies, endemic to 97 countries in parts of the tropics, subtropics, and in Southern Europe [1]. An estimated 700,000 to 1 million new cases of leishmaniases and 20,000–30,000 deaths occur annually. In 2017, 94% of new cases reported to the WHO occurred in seven countries: Brazil, Ethiopia, India, Kenya, Somalia, South Sudan, and Sudan [2]. The three main clinical syndromes of leishmaniasis include cutaneous leishmaniasis (CL), muco-cutaneous leishmaniasis (MCL), and visceral leishmaniasis (VL) [3].

Leishmaniasis is one of the most neglected diseases in terms of drug development [4,5]. Without any effective vaccine against leishmaniasis in humans at present, the control of these parasites relies solely on chemotherapy [6]. The current chemotherapies suffer from several drawbacks due to toxicity, long-term treatment, severe adverse effects, high cost, invasive administration routes, low effectiveness, and commonly, drug resistance [4,7,8]. Therefore, there is an urgent need for novel compounds with substantial antileishmanial effect and low toxicity to the host.

*Leishmania* and all members of the *Trypanosomatidae* family depend on polyamines (PA) for growth and survival [9,10]. The first enzyme involved in PA biosynthesis that hydrolyses arginine into ornithine and urea is arginase [11]. Arginase inhibition can lead to oxidative stress in parasite cells owing to a deficiency in trypanothione production that neutralizes reactive macrophage-derived oxygen and nitrogen species [12], consequently leading to infection control [13]. Arginase is important for parasite survival: deleting the arginase gene is auxotrophic for polyamines [14]. Being arginase distinct from the mammalian target [11] and absolute necessity of the enzyme for survival of the pathogen, it represents a key parasite drug target [15]. 

Oxadiazoles have been reported as potent anti-leishmanial agents [16,17]. Potent antimicrobial and anti-leishmanial activity are also documented for indolizine-containing compounds [18,19]. The MetIDB database was screened in previous work with in silico procedures and ten promising flavonoids were proposed as anti-leishmanial candidates [20]. In this study, a series of novel oxadiazoles and indolizine containing compounds were synthesized, screened in silico with the Electron Ion Interaction Potential/Average Quasi Valence Number (EIIP/AQVN) filter, followed by ligand-based virtual screening and molecular docking to *Leishmania* arginase. In addition, the top hits were virtually screened against human arginase and anti-targets. The best selected candidates for anti-leishmanial compounds were subjected to experimental testing. The experimental results show measurable in vitro anti-leishmanial activity for three compounds.

## 2. Results

### 2.1. Chemistry

#### 2.1.1. Synthesis of 1,2,4-Oxadiazoles

*N*-acylhydrazones are important and versatile scaffolds. They can be obtained by conventional [21] or non-conventional methods [22,23]. They can be considered either as final products with important biological properties or stable intermediates in the design and construction of other motifs like 1,2,4-oxadiazoles. In our hands, as *N*-acylhydrazones **1**–**5** were prepared on an automated platform, we conducted synthesis by refluxing an equimolar mixture of hydrazide and aldehyde in ethanol with a catalytic amount of hydrochloric acid (Scheme 1). Purifications by filtration of the solid compounds were parallelized and we were able to isolate about 200 compounds in four batches. The five compounds used in this work are shown in Table 1.

1,2,4-Oxadiazoles can be considered as one of the most important 5-membered heteroaromatic rings found in many pharmaceutical compounds. Among the various synthetic approaches reported in the literature [24], one concerns reaction under conventional or non-conventional methods of amidoximes with suitably activated acid derivatives [25].

Synthesis of the 3-[3,4-(methylenedioxy)phenyl)]-5-(methoxymethyl)-1,2,4-oxadiazole **8** was achieved in two steps and 90% total yield by the reaction of 3,4-(methylenedioxy)benzonitrile with hydroxylamine to afford quantitatively amidoxime **6** followed by the reaction with methoxyacetyl chloride under microwave irradiation (Scheme 2). This compound was prepared during a small molecule library synthesis program, and the protocols established to allow the automation and the parallelization of reactions. The first step of the procedure was applied to 3-nitrobenzonitrile, affording also in quantitative yield the corresponding 3-nitrobenzamidoxime.

#### 2.1.2. Synthesis of Indolizines

The 1,3-dipolar cycloaddition reaction of azomethine ylide containing nitrogen in a 6-membered ring such as pyridine and isoquinoline was applied for the synthesis of the target compounds. In that respect, 1-((5-nitro-1*H*-benzo[d]imidazol-2-yl)methyl)pyridin-1-ium chloride **9** and 2-((5-nitro-1*H*-benzo[d]imidazol-2-yl)methyl)isoquinolin-2-ium chloride **10**(2), (Scheme 3), which were prepared from the reaction of pyridine/isoquinoline with 2-(chloromethyl)-5-nitro-1*H*-benzo[d]imidazole (Scheme 3), as described previously [26,27], undergo a dehydrohalogenation reaction in the presence of triethylamine in chloroform to give the corresponding pyridinium **9**/isoquinolinium **10** ylide in 75% and 95% yield, respectively. The reaction of the latter (Scheme 4) with DMAD or methyl propiolate at room temperature and overnight afforded, after workup and column chromatography, the indolizine **11** (25% yield), and benzindolizines **12** and **13** (28% and 27%, respectively). The obtained compounds were fully characterized by ^1^H, ^13^C NMR, mass spectra, and IR.

### 2.2. Virtual Screening

#### 2.2.1. EIIP Filtering

We used the previously developed average quasi valence number/electron ion interaction potential (EIIP/AQVN) criteria for selection of *Leishmania* arginase compounds [20].

#### 2.2.2. 3D QSAR Filtering

39 molecules were obtained after filtering with EIIP criteria and were subjected to prediction of their activity using the above described arginase 3D-quantitative structure-activity relationship (3D-QSAR) model. The two criteria for selection were: (1) partial least square (PLS) scores in the vicinity of compounds from the model (Figure 1); and (2) best ranking by predicted IC_50_ values. This filtering gave 10 candidates that were used for docking into the *Leishmania* arginase structure model, human arginase crystal structure, and off-target (anti-target) affinity calculation.

#### 2.2.3. Arginase Docking

Ten compounds were docked into both the parasite arginase model structure and the crystal structure of human arginase. Docking scores of the best-docked conformations, along with experimental measurements are presented in Table 2. Although PLS prediction and docking score results were promising, the experiments showed significant activity of six compounds. However, due to their toxicity, only one candidate, compound **2**, is acceptable thanks to its selectivity index (SI, CC_50_ for macrophage/IC_50_ for promastigotes) ≥ 2, with an IC_50_ value on intramacrophage amastigotes < 5 µM. The highest ranked docking conformation of this compound in the *Leishmania* arginase model is presented in Figure 2.

#### 2.2.4. Anti-Target Interaction Matrix

The top eight compounds from the filtering were then assessed against the anti-target battery. The results of the docking of all the final compounds against the battery of five anti-targets are shown in Figure 3 (full table of docking scores Table 3).

There was broad general agreement between the five anti-targets. None of the compound’s docking score surpassed that of the threshold for CYP P450 2a6, 2c9, or 3a4, which may be an indication of the relative size of the ligands. There were more interactions found with the anti-targets PXR and SULT. Compounds **3** and **5** had the highest combined score of 1.0 for all anti-targets combined, while most compounds had even lower interactions.

All of the compounds had zero PAINS flags, passed Lipinksi’s rule-of-five, and were predicted to be soluble or moderately soluble, have high gastrointestinal absorption, and potential (using a support-vector-machine) CYP 2a6 binding for **2**, **3**, **4**, **5**, and CYP 3a4 for **8**, according to filters [28].

#### 2.2.5. In vitro Evaluation of Anti-Leishmanial Activity

Finally, we selected eight hit compounds for experimental testing. Since isolated arginase was not available, the selected compounds were assayed for their in vitro inhibition activity against axenic amastigote and intramacrophage amastigote forms of *Leishmania donovani* (Table 3). A broad range of activities against axenic and intramacrophage amastigotes forms of *Leishmania donovani* was found with IC_50_ values in the range between 1–2 and 55–61 µM. The compounds exhibited similar activities on both axenic amastigotes and intramacrophage amastigotes. The most potent inhibitors were also slightly toxic with CC_50_ values in the range from 4 to 16 µM. 

Regarding structure–activity relationships, the replacement of a thiophene group (compound **4**) by a pyridine group (compound **5**) was responsible for a loss of activity by a factor of five and also a strong reduction of cytotoxicity. Among the tested compounds, one compound, **2**, is the most active despite having a not negligible toxicity. Its chemical structure has two thiophene groups. The replacement of a thiophene group (compound **2**) by a furan group (compound **3**) led to a total loss of activity and cytotoxicity. The replacement of the bromothiophene group (compound **2**) by a benzothiazole group (compound **4**) was responsible for a five-fold reduction of the anti-leishmanial activity and four-fold reduction of cytotoxicity. The replacement of bromothiophene (compound **3**) by a methylbenzothiazole group (compound **1**) enhanced both the antileishmanial activity and cytotoxicity. In the phenotypic based screening, the most promising compound **2** (Table 3) has an IC_50_ value of 2.18 µM, 20 times higher than that of amphotericin B, the reference compound. Regarding the selectivity index (SI) values, compound **2** has an SI of 2, whereas amphotericin B had an SI of 48 against *Leishmania* axenic amastigotes. It is also less than four times higher than the reference drug miltefosine (0.31 µM) [29]. The obtained in vitro results clearly require additional pharmacomodulations to reduce the cytotoxicity and enhance the antileishmanial activity in order to achieve better selectivity index values. 

## 3. Discussion

Inhibition of enzymes of the polyamine–trypanothione metabolism including arginase is considered as one of the best options for the treatment of *Leishmania* since many of these enzymes passed both target validation and chemical validation [10]. In the quest for potential treatment options against leishmaniasis, increasing interest has been shown for *N*-acylhydrazone, oxadiazole, and indolizine containing compounds. These types of compounds targeting arginase represent an interesting strategy in the search for a new anti-leishmanial treatment.

In our hands, *N*-acylhydrazones were prepared on an automated platform where reaction conditions and purifications by filtration of the solid compounds were parallelized for 50 compounds in each run (about 200 compounds synthetized in four batches). In that respect, reaction conditions were generalized, and for some of the compounds, yields are not optimum. The same is also true for the 1,2,4-oxadiazoles where the reaction conditions were chosen in order to prepare a small-molecule library (150 compounds synthetized). The reaction conditions first afforded the amidoxime compounds quantitatively, then the reaction with activated acid furnished in good yield the 1,2,4 oxadiazoles, where one example is used in the current study. Concerning the indolizine compounds, the azomethine ylide prepared in situ by 1,3 dipolar cycloaddition was chosen. The yields obtained for compounds **11**–**13** are poor, varying between 25–28% after careful purification. The insolubility of the chloride salts could be the reason for the low yields. As one of the research programs of the group concerns 1,3 dipolar cycloadditions using this type of ylides, efforts are currently oriented to the synthesis of indolizine-bearing derivatives under non-conventional methods in order to obtain focused small libraries of these compounds.

The compounds thus obtained for this study were subsequently screened. The ten selected compounds from our study with favorable in silico interaction profiles with arginase and against the anti-targets were favorable for further experimental testing and represent better initial points than screening compounds at random. Three compounds were found to be active against *Leishmania donovani*. One product, compound **2**, is better than the others as an interesting molecular template for further development of new anti-leishmanial agents due to its observed experimental activity against parasite amastigotes, better selectivity index, and predicted interactions with targets and anti-targets. Interestingly, the compound that had the best selectivity index, the most promising compound **2**, had a docking score that was approximately 0.8 kcal/mol stronger for the parasite arginase than for human arginase. This may also indicate that a larger degree of selectivity for a compound may be picked up by the procedure carried out here.

Considering the mechanism of action of both chemical series, the results obtained in this paper prompt us to develop further pharmacomodulations in order to diminish the cytotoxicity and enhance the anti-leishmanial activity. In any case, despite the level of cytotoxicity of compound **2** being in a similar range to that of the reference compound AmB, the obtained results justify the determination of the maximal tolerated dose in mice, and then, the in vivo evaluation of compound **2** on the *L. donovani*/BALB/c mice model. 

## 4. Materials and Methods 

### 4.1. Chemistry and Physico-Chemical Analyses

Melting points (m.p.) were determined using a Mettler Toledo MP50 system and were uncorrected. ^1^H and ^13^C NMR spectra were recorded in CDCl_3_ and/or DMSO-d6 using a Bruker AC 300 (^1^H) or 75 MHz (^13^C) instruments, except for compounds **1**, **2**, **4**, **6**, and **7**: ^13^C-NMR analyses were conducted with a Bruker AVANCE500 at 298 K (125.75 MHz). Chemical shifts are given in δ parts per million (ppm) and referenced to external TMS. 

Automated syntheses were carried out on an Accelerator SLT-106 workstation from Chemspeed and microwave assisted reactions on a SWave workstation from Chemspeed equipped with a Biotage Initiator reactor. 

High-resolution mass spectra (MS) were recorded on a GCT Premier Mass Spectrometer (Waters, MA, USA). Liquid Chromatography/Mass Spectrometry (LC/MS) analyses were performed on an Autopurif system from Waters (PDA 2996, MS 3100, Pump 2545, MA, USA) with a Gemini-NX column (5, C18, 110A, 50 × 4.6 mm) from Phenomenex. Analyses were done with 1 mL/min flow and a gradient of water/acetonitrile containing 0.05% of formic acid (0.0 to 1.0 min: 90/10; 1.0 to 5.0 min: 90/10 to 0/100; 5.0 to 6.5 min: 0/100; 6.5 to 7.0 min: 0/100 to 90/10; 7.0 to 12.0: 90/10) and purities were determined at 254 nm.

#### 4.1.1. General Procedure for the Automated Synthesis of Compounds **1**–**5**:

Aldehyde (1.00 mmol), hydrazide (1.00 mmol), ethanol (4.0 mL) and 50 μL of aqueous 1M HCl were added in a 13 mL double jacket reactor from Chemspeed equipped with condenser. The mixture was shaken at 700 rpm for 4 h at 80 °C, then cooled down to 20 °C. The precipitate was filtrated and successively washed with 2 mL of ethanol and 2 mL of ethyl ether to obtain the product.



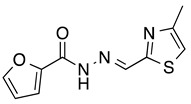



*N’-[(4-Methyl-2-thiazolyl)methylene]furan-2-carbohydrazide* (**1**): Tan solid, yield: 61%, m.p.: 121–124 °C. LC purity (254 nm): 92%. ^1^H-NMR (300 MHz, DMSO) δ ppm: 12.22 (s, 1H), 8.62 (s, 1H), 8.00–7.93 (m, 1H), 7.41–7.33 (m, 2H), 6.73 (dd, *J* = 3.3, 1.5 Hz, 1H), 2.39 (d, *J* = 0.9 Hz, 3H). ^13^C-NMR (125.75 MHz, DMSO) δ ppm: 163.72, 157.80, 154.60, 153.93, 142.57, 116.86, 116.21, 115.16, 112.74, 112.38, 17.14. HRMS (DCI-CH_4_, TOF) *m*/*z*: [M + H]^+^ calc. for C_10_H_10_N_3_O_2_S: 236.0494. Found: 236.0494.



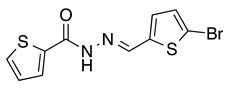



*N’-[(5-Bromo-2-thiophenyl)methylene]thiophene-2-carbohydrazide* (**2**): Light yellow solid, yield: 72%, m.p.: 195–198 °C. LC purity (254 nm): 99%. ^1^H-NMR (300 MHz, DMSO) δ ppm: 11.89 (s, 1H), 8.58 and 8.20 (2 broad singlets, 1H), 8.05–7.93 (m, 1 H), 7.88 (br, 1H), 7.33 (dd, *J* = 4.5, 0.6 Hz, 1H), 7,27 (d, *J* = 3.9 Hz, 1H), 7,22 (dd, *J* = 5.1, 3.9 Hz, 1H). ^13^C-NMR (125.75 MHz, DMSO) δ ppm: 161.58, 158.40, 142.42, 141.41, 141.24, 138.50, 138.15, 135.65, 135.26, 133.46, 132.47, 131.98, 129.51, 128.60, 127.14, 115.25. HRMS (ES, TOF) *m*/*z*: [M + H]^+^ calc. for C_10_H_8_N_2_OS_2_Br: 314.9261. Found: 314.9250.



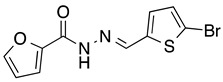



*N’-[(5-Bromo-2-thiophenyl)methylene]furan-2-carbohydrazide* (**3**): Light yellow solid, yield: 71%, m.p.: 201–202 °C. LC purity (254 nm): 99%. ^1^H-NMR (300 MHz, DMSO) δ ppm: 11.89 (s, 1H), 8.57 (br, 1H), 7.94 (s, 1H), 7.32–7.24 (m, 3H), 6.70 (s, 1H). ^13^C-NMR (75 MHz, DMSO) δ ppm: 154.1, 146.5, 146.0, 142.1, 140.9, 131.5, 131.3, 115.1, 114.8, 112.1. HRMS (ES, TOF) *m*/*z*: [M + H]^+^ calc. for C_10_H_8_N_2_O_2_SBr: 298.9490. Found: 298.9492.



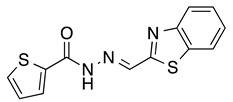



*N’-[(2-Benzothiazolyl)methylene]thiophene-2-carbohydrazide* (**4**): Light yellow solid, yield: quant., m.p.: 231–234 °C. LC purity (254 nm): 99%. ^1^H-NMR (300 MHz, DMSO) δ ppm: 12.37 (s, 1H), 8.74 and 8.42 (2 br, 1H), 8.20-8.12 (m, 1H), 8.12–7.87 (m, 3H), 7.55 (ddd, *J* = 14.7, 7.5, 1.5 Hz, 1H), 7.50 (ddd, *J* = 15.9, 7.5, 1.5 Hz, 1H), 7.27 (dd, *J* = 4.8, 3.6 Hz, 1H). ^13^C-NMR (125.75 MHz, DMSO) δ ppm: 165.52, 164.86, 161.86, 158.45, 153.62, 142.24, 138.56, 135.87, 134.63, 130.34, 128.77, 127.17, 127.04, 123.76, 123.00. HRMS (ES, TOF) *m*/*z*: [M + H]^+^ calc. for C_13_H_10_N_3_OS_2_: 288.0265. Found: 288.0277.



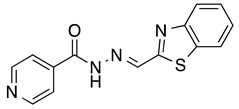



*N’-[(2-Benzothiazolyl)methylene]pyridine-4-carbohydrazide* (**5**): Off-white solid, yield: 70%, m.p.: 234–237 °C. LC purity (254 nm): 92%. ^1^H-NMR (300 MHz, DMSO) δ ppm: 12.64 (s, 1H), 8.88–8.81 (m, 2H), 8.79 (s, 1H), 8.21–8.11 (m, 1H), 8.11–8.03 (m, 1H), 7.92–7.82 (m, 2H), 7.61–7.47 (m, 2H). ^13^C-NMR (75 MHz, DMSO) δ ppm: 164.7, 162.0, 153.1, 150.3, 143.3, 140.1, 134.1, 126.8, 126.7, 123.4, 122.6, 121.7. HRMS (ES, TOF) *m*/*z*: [M + H]^+^ calc. for C_14_H_11_N_4_OS: 283.0654. Found: 283.0656.

#### 4.1.2. General Procedure for the Automated Synthesis of Compounds **6** and **7**:

Nitrile (20.00 mmol), ethanol (30.0 mL) and hydroxylamine (50% in water, 2.0 mL) were added in a 75 mL double jacket reactor from Chemspeed equipped with condenser. The mixture was shaken at 600 rpm for 5 h at 80 °C, then cooled down to 20 °C. The reaction medium was evaporated under reduced pressure and the solid was finally dried overnight in a heated (40 °C) vacuum oven.



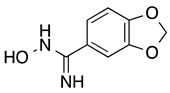



*3,4-(Methylenedioxy)benzamidoxime* (**6**): White solid, yield: quant. ^1^H-NMR (300 MHz, DMSO) δ ppm: 9.51 (s, 1H), 7.22–7.16 (m, 2H), 6.93–6.88 (m, 1H), 6.03 (s, 2H), 5.72 (s, 2H). ^13^C-NMR (125.75 MHz, DMSO) δ ppm: 150.98, 148.25, 147.56, 127.87, 119.75, 108.30, 106.17, 101.61.



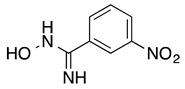



*3-Nitrobenzamidoxime* (**7**): Yellow solid, yield: quant., m.p.: 182-184 °C. ^1^H-NMR (300 MHz, DMSO) δ ppm: 9.97 (s, 1H), 8.51 (m, *J* = 2.1 Hz, 1H), 8.23 (ddd, *J* = 8.1, 2.4, 0.9 Hz, 1H), 8.12 (ddd, *J* = 7.8, 1.5, 1.2 Hz, 1H), 7,68 (t, *J* = 8.1 Hz, 1H), 6.10 (s, 2H). ^13^C-NMR (125.75 MHz, DMSO) δ ppm: 149.60, 148.22, 135.40, 132.04, 130.24, 123.93, 120.40. HRMS (DCI-CH_4_, TOF) *m*/*z*: [M + H]^+^ calc. for C_7_H_8_N_3_O_3_: 182.0566. Found: 182.0559.

#### 4.1.3. Procedure for the Synthesis of Compound **8**:

A mixture of 3,4-(methylenedioxy)benzamidoxime (**6**, 180 mg, 1.00 mmol), acetonitrile (3.0 mL), thiethylamine (153 μL, 1.10 mmol) and methoxyacetyl chloride (91 μL, 1.00 mmol) was placed in a sealed vial and exposed to microwave irradiation (130 °C, 1 h). The reaction medium was evaporated under reduced pressure and the solid was washed with 15 mL of ethyl acetate to give **8** (188 mg, 80% yield).



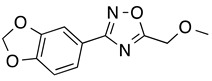



*3-[3,4-(Methylenedioxy)phenyl)]-5-(methoxymethyl)-1,2,4-oxadiazole* (**8**): Light tan solid, m.p.: 56–57 °C. LC purity (254 nm): 98%. ^1^H-NMR (300 MHz, DMSO) δ ppm: 7.58 (dd, *J* = 8.1, 1.8 Hz, 1H), 7.45 (dd, *J* = 1.5, 0.3 Hz, 1H), 7.09 (dd, *J* = 8.1, 0.3 Hz, 1H), 6.14 (s, 2H), 4.80 (s, 2H), 3.42 (s, 3H). ^13^C-NMR (75 MHz, DMSO) δ ppm: 176.5, 167.2, 150.1, 148.0, 122.1, 119.6, 109.0, 106.6, 101.9, 64.4, 58.8. HRMS (DCI-CH_4_, TOF) *m*/*z*: [M + H]^+^ calc. for C_11_H_11_N_2_O_4_: 235.0719. Found: 235.0712.

#### 4.1.4. General Procedure for the Synthesis of Compounds **11**–**13**:

Compounds 1-((5-nitro-1*H*-benzo[d]imidazol-2-yl)methyl)pyridin-1-ium chloride **9** and 2-((5-nitro-1*H*-benzo[d]imidazol-2-yl)methyl)isoquinolin-2-ium chloride **10** were prepared as previously reported and gave the same experimental characteristics. A mixture of pyridinium/ isoquinolinium salt (1 mmol, 1.0 eq.) and alkyne (1.1 mmol, 1.1 eq.) in chloroform was stirred at 0 °C. Triethylamine (1.3 mmol, 1.3 eq.) was added dropwise and the mixture was further stirred at room temperature for 24 h. The reaction medium was evaporated under reduced pressure and the obtained residue was purified by flash chromatography.

99 mg of compound **11** were obtained (25% yield).



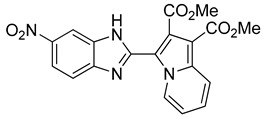



*Dimethyl 3-(5-nitro-1H-benzo[d]imidazol-2-yl)indolizine-1,2-dicarboxylate* (**11**): m.p.: 234 °C. R_f_: 0.6 EP/EtOAc (7:3 *v*/*v*). ^1^H-NMR (300 MHz, CDCl_3_) δ ppm: 11.65 (s, 1H), 10.30 (d, *J* = 7.2, Hz, H5), 8.72 (d, *J* = 2.1 Hz, H4′), 8.32–8.18 (m, H6′ + H8), 7.59 (d, *J* = 8.8 Hz, H7′), 7.38 (dd, *J* = 9.0, 6.7 Hz, H7), 7.11 (dd, *J* = 7.2, 6.7 Hz, H6), 4.08 (s, 3CH_3_), 4.00 (s, 3CH_3_). ^13^C-NMR (75 MHz, DMSO) δ ppm: 169.2, 163.7, 148.2, 147.5, 147.3, 143.9, 142.6, 137.0, 173.0, 131.9, 128.4, 126.1, 119.3, 119.0, 115.2, 110.9, 107.9, 53.6, 51.8. HRMS (ES, TOF) *m*/*z*: [M + H]^+^ calc. for C_19_H_14_N_4_O_6_: 395.0992. Found: 395.0988. IR (ATR) (cm^−1^): 3347, 3117, 2954, 1716, 1702, 1503, 1215, 749.

125 mg of compound **12** were obtained (28% yield).



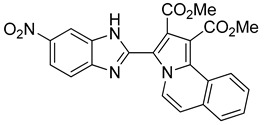



*Dimethyl 3-(5-nitro-1H-benzo[d]imidazol-2-yl)pyrrolo[2,1-a]isoquinoline-1,2-dicarboxylate* (**12**): m.p.:162 °C. R_f_: 0.6 EP/THF (1:4 *v*/*v*). ^1^H-NMR (300 MHz, CDCl_3_) δ ppm: 11.75 (s, NH), 8.98 (d, *J* = 7.5 Hz, H5), 8.59 (s, H4′), 8.35–8.26 (m, H10), 8.17 (d, *J* = 8.9 Hz, H6′), 7.86-7.83 (m, H7+H7′), 7.65-7.62 (m, H8+H9), 7.33 (d, *J* = 7.5 Hz, H6), 4.00 (s, CH_3_), 3.85 (s, CH_3_). ^13^C-NMR (75 MHz, DMSO) δ ppm: 166.9 (CO), 164.1 (CO), 146.9 (C5′), 143.4 (C2), 129.0 (CH8 + CH9), 128.9 (C11), 128.7 (C), 128.2 (CH7), 123.9 (C), 123.9 (CH10), 123.8 (CH5), 119.3 (C), 118.7 (CH6′), 117.5 (C3), 115.4 (CH7′), 115.3 (CH6), 113.1 (CH4′), 111.0 (C), 53.3 (CH_3_), 53.0 (CH_3_). HRMS (ES, TOF) *m*/*z*: [M + H]^+^ calc. for C_23_H_17_N_4_O_6_: 445.1148. Found: 445.1127. IR (ATR) (cm^−1^): 3399, 3109, 2990, 1727, 1686, 1519, 1340, 1210, 734.

195 mg of compound **13** were obtained (27%)



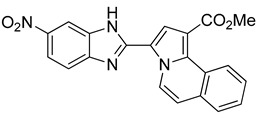



*Methyl 3-(5-nitro-1H-benzo[d]imidazol-2-yl)pyrrolo[2,1-a]isoquinoline-1-carboxylate* (**13**): m.p.: 280 °C. R_f_: 0.4 EP/EtOAc (4:1 *v*/*v*). ^1^H-NMR (300 MHz, CDCl_3_) δ ppm: 13.60 (s, NH), 10.03 (d, *J* = 7.5 Hz, H5), 9.78–9.73 (m, H4′), 8.54–8.51 (m, H10), 8.23 (s, H2), 8.14–8.08 (m, H6′), 7.93–7.89 (m, H7′), 7.70–7.63 (m, H7,H8,H9), 7.52 (d, *J* = 7.5 Hz, H6), 3.93 (s, CH_3_). ^13^C-NMR (75 MHz, DMSO) δ ppm: 164.7, 143.1, 139.5, 131.7, 129.5, 129.2, 128.0, 127.5, 127.2, 127.0, 126.1, 125.2, 124.7, 124.5, 120.3, 119.0, 116.4, 115.4, 115.1, 52.1. HRMS (ES, TOF) *m*/*z*: [M + H]^+^ calc. for C_21_H_24_N_4_O_4_: 387.1067. Found: 387.1070. IR (ATR) (cm^−1^): 2920, 2851, 1708, 1521, 1341, 1202, 760.

### 4.2. Ligands

The source for data was internal libraries of synthetized oxadiazoles and indolizine-containing compounds.

### 4.3. Virtual Screening

Starting with a filter based on EIIP/AQVN values and subsequently, 3D QSAR filtering, and arginase docking, hit compounds were docked into anti-targets involved in the metabolism of compounds in order to tag their possible interactions.

#### 4.3.1. EIIP/AQVN

AQVN and the EIIP can give indication on long-range biomolecule interaction (over 0.5 nm) [30], based on the general model pseudopotential [31]
EIIP = 0.25 Z* sin(1.04 π Z*)/2π,(1)
where Z* is the average quasi-valence number (AQVN):
Z* = ∑m(n*i* Z*i* / N),(2)
where Z*i* is the valence number of the *i*th atomic component, n*i* is the number of atoms of the *i*th component, m is the number of atomic components in the molecule, and N is the total number of atoms. EIIP values are calculated according to equations 1 and 2 are expressed in Rydberg units (Ry).

Similarity in AQVN and EIIP values may give a clue on a common therapeutic target, thus setting up criteria for virtual screening of molecular libraries for compounds with similar therapeutic properties [20].

#### 4.3.2. 3D QSAR

We used our previously reported 3D QSAR model of *Leishmania amazonenzis* [20].

### 4.4. Arginase

The crystal structure of *Leishmania mexicana* with ABH (4iu0.pdb) was used to generate a model for *L. amazoniensis*. Docking was performed with Autodock 4.6.2. The Lamarckian GA method was used to perform the conformational search.

### 4.5. Anti-Targets

The docking for the anti-targets was carried out with Glide XP [32] and proteins with resolution ≤ 2.6 Å (1m13, 2a3r, 1z10, 1og5, 1tqn) [33,34]. The thresholds for determining a strong interaction were set to −7.7, −6.3, −7.6, −8.7, and −7.5 kcal/mol. An estimate of the binding was changed to interaction codes:
Score = 0.0 if ΔG − ΔGref > 0.5,(3)
Score = 0.5 if |ΔG − ΔGref| ≤ 0.5,(4)
Score = 1.0 if ΔG − ΔGref < −0.5,(5)
where ΔGref is the docking score of the protein with co-crystallized active reference ligand, and ΔG is the docking score for a ligand bound to that protein binding site. Shades were then set to white, grey, and black.

### 4.6. Biological Evaluation

#### 4.6.1. Cultures for Parasites and Cells

*Leishmania donovani* promastigotes (MHOM/ET/67/HU3/LV9) were cultured with 5% CO_2_ in M199 complete medium containing M199 medium supplemented with 100 μM adenosine, 0.5 mg/L hemin, 40 mM Hepes pH 7.4 and 10% heat inactivated fetal bovine serum (HIFBS) in the dark at 26 °C. Late log promastigotes diluted at 1 × 10^6^/mL in M199 complete medium acidified at pH 5.5 and cultured at 37 °C with 5% CO_2_ provided the cultures of axenic amastigotes of *L. donovani*.

#### 4.6.2. Macrophage

RAW 264.7 macrophages were grown with 5% CO_2_ in DMEM complete medium containing Dulbecco’s Modified Eagle’s Medium (DMEM) with 100 U/mL penicillin-streptomycin, and 10% HIFBS at 37 °C.

#### 4.6.3. In Vitro Anti-Leishmanial Compound Testing on Axenic and Intramacrophage Amastigotes:

Previously described protocols [35] were adapted for evaluation of compounds on *L. donovani*. For axenic amastigotes, two-fold serial dilutions of the compounds were done in 96-well microplates with 100 μL of complete medium (see above). Axenic amastigotes were then added to each well at a density of 106 mL in a 200 μL final volume. After 72 h of incubation with 5% CO_2_ at 37 °C, 20 μL of resazurin (450 μM) was added to each well and further incubated with 5% CO_2_ at 37 °C for 24 h in the dark. Resazurin is reduced to resorufin in living cells and can be then monitored by measuring OD570nm (resorufin) and OD600nm (resazurin; Lab systems Multiskan MS). Compound activity was expressed as IC_50_ (µM). The drug used for reference was Amphotericin B (AmB).

For intramacrophage amastigote testing, RAW 264.7 macrophages were incubated with 5% CO_2_ at 37 °C for 24 h after being plated in 96-well microplates at a density of 2 × 10^4^ cells per well. Axenic amastigotes were differentiated as described above, centrifuged at 2000 g for 10 min, re-suspended in DMEM complete medium, and added to each well to reach a 16:1 parasite to macrophage ratio. 24 h of infection with 5% CO_2_ at 37 °C proceeded, and extracellular parasites were removed. This was followed by addition to each well of DMEM complete medium (100 μL) containing two-fold serial dilutions of the compounds from a maximal concentration of 100 μM. The medium was removed following treatment of 48 h and replaced by Direct PCR Lysis Reagent (100 μL; Euromedex) before 3 freeze-thaw cycles at room temperature, addition of 50 μg/mL proteinase K, and a final incubation at 55 °C overnight to allow cell lysis. 10 μL of each cell extract was then added to 40 μL of Direct PCR Lysis reagent containing Sybr Green I (0.05%; Invitrogen). Mastercycler^®^ realplex (Eppendorf) was used to monitor DNA fluorescence. Compound activity was expressed as IC_50_ (µM). The drug used for reference was Amphotericin B (AmB).

#### 4.6.4. Cytotoxicity Tests

RAW 264.7 macrophages were used for cytotoxicity testing. Cells were plated in 96-well microplates at a density of 2 × 10^4^ cells per well. Twenty-four hours of incubation with 5% CO_2_ at 37 °C followed, and then the medium was removed and to each well, 100 μL of DMEM complete medium containing two-fold serial dilutions of the compounds were added. This was followed by 48 h of incubation with 5% CO_2_ at 37 °C, after which 10 μL of resazurin (450 μM) was added to each well, and further incubated with 5% CO_2_, in the dark for 4 h at 37 °C. Cell viability was then monitored as above. Compound cytotoxicity was expressed as CC_50_ (Cytotoxic Concentration 50%: concentration inhibiting the macrophages growth by 50%).

## 5. Conclusions

There is a growing and urgent need for new chemical compounds for the treatment of several leishmaniases given the lack of vaccines and the side-effects and lack of efficacy of presently-used chemicals. Four different series of compounds were synthetized and tested as potent anti-leishmanial compounds. While the indolizine compound does not present good activity, and the aldoxime and 1,2,4-oxadiazole tested are both equally moderately active, *N*-acylhydrazones presented much better activities. Among them, compound **2** is better than the others and could be considered as an interesting molecular template for further development of new anti-leishmanial agents due to its observed experimental activity against parasite *Leishmania donovani* axenic and intramacrophage amastigotes, better selectivity index, and predicted interactions with targets and anti-targets. It has a selectivity index of 2.2. Further work may be carried out to improve the toxicity and potency profiles of these compounds.

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
