# Peer review of "Synthesis, In Silico, and In Vitro Evaluation of Anti-Leishmanial Activity of Oxadiazoles and Indolizine Containing Compounds Flagged against Anti-Targets"

_molecules, 2019, doi:10.3390/molecules24071282_

Reviewer 1 Report

Leishmaniasis is a neglected disease  caused by protozoan parasites belonging to the genus Leishmania. It is endemic to 97 countries worldwide. It is estimated that 700,000 to 1 million new cases of the infection occur annually, generating 20,000–30,000 deaths. In terms of drug development against leishmaniasis, there is an urgent need for novel compounds with good leishmanicidal activity and low toxicity to the hosts.

Based on some characteristics of arginase (such as its dissimilarity to the mammalian target and its critical role for leishmania survival), many studies support the idea that this enzyme is as a key parasite drug target.

The authors described herein the synthesis of a series of novel oxadiazoles and indolizine containing compounds. Those compounds were screened in silico using an EIIP/AQVN filter followed by ligand-based virtual screening and molecular docking. In addition, the selected hits were screened versus human arginase as well as against an anti-target battery to tag putative interactions with these essential proteins critical for the metabolism.

Their experimental results showed three of those compounds exhibited in vitro leishmanicidal activity. Interestingly, one compound was active against intramacrophage amastigotes with an IC50 value around 2 μM and a selectivity index of 2.2 (a not negligible toxicity).

This work represents a novel and interesting study. See below some few comments that may improve this proof-of-concept study.

MINOR COMMENTS:

1. Table 2: Please, improve the headings.

2. How conserved is Arginase in Trypanosomatids? Regarding compound 2, what activity do you expect in other species of Leishmania (for example, L. major, L. infantum)?

3. The evaluation of coumpounds cytoxicity was performed on RAW 264.7 macrophages. It is well known that RAW 264.7 macrophages are immortalized cell lines. Therefore, they could exhibit less sensitivity to some treatments or drug resistance. The use of peritoneal macrophages or bone marrow-derived macrophages could be good and valuable alternatives for further validation assays.

Author Response

Reviewer 1:

Leishmaniasis is a neglected disease caused by protozoan parasites belonging to the genus Leishmania. It is endemic to 97 countries worldwide. It is estimated that 700,000 to 1 million new cases of the infection occur annually, generating 20,000–30,000 deaths. In terms of drug development against leishmaniasis, there is an urgent need for novel compounds with good leishmanicidal activity and low toxicity to the hosts.

Based on some characteristics of arginase (such as its dissimilarity to the mammalian target and its critical role for leishmania survival), many studies support the idea that this enzyme is as a key parasite drug target.

The authors described herein the synthesis of a series of novel oxadiazoles and indolizine containing compounds. Those compounds were screened in silico using an EIIP/AQVN filter followed by ligand-based virtual screening and molecular docking. In addition, the selected hits were screened versus human arginase as well as against an anti-target battery to tag putative interactions with these essential proteins critical for the metabolism.

Their experimental results showed three of those compounds exhibited in vitro leishmanicidal activity. Interestingly, one compound was active against intramacrophage amastigotes with an IC50 value around 2 μM and a selectivity index of 2.2 (a not negligible toxicity).

This work represents a novel and interesting study. See below some few comments that may improve this proof-of-concept study.

MINOR COMMENTS:

R1.1: Table 2: Please, improve the headings.

A1.1: The authors thank the Reviewer for their comments. We have now improved the headings of Table 2 in the manuscript text

R1.2:  How conserved is Arginase in Trypanosomatids? Regarding compound 2, what activity do you expect in other species of Leishmania (for example, L. major, L. infantum)?

A1.2: A BLAST search using the L. mexicana structure that we used returns high similarity (over 95%) with other Trypanosoma organisms, such as Leishmania donovani, L. Amazonensis, L. infantum, L. major, L. Tropica, L. tarentolae, L. braziliensis. L. panamensis, and around 60% with Leptomonas seymouri, Crithidia acanthocephali, Leptomonas pyrrhocoris, Strigomonas galati, Angomonas desouzai, Angomonas deanei, Strigomonas culicis, Strigomonas oncopelti, as well as several yeasts, molds, and fungi. Trypanosoma brucei has a very different structure (crystal structure of an arginase-like protein shows only one conserved amino-acid for binding the catalytic ion. (See Y. Hai, DOI: 10.1021/bi501366a Biochemistry 2015, 54, 458−471)) 

Arginase activity has not been found in all Trypanosomatids. Thus, it has been detected in Leishmania, Crithidia and Leptomonas but not in Trypanosoma, Herpetomonas or Phytomonas. 

The total amino acid sequences homology within Leishmania spp. (L.  mexicana, L. infantum, L. major, and L. tropica) ARGs is always more than 95%. Residues essential for L-arginine as a substrate and inhibitors binding are highly conserved between all these ARGs (Badirzadeh et al., PLoS Negl. Trop. Dis., 1-22, 2017. https://doi.org/10.1371/journal.pntd.0005774 

Regarding the activity of compound 2 on other Leishmania species, we expect similar activities as those observed in L. donovani, but considering the polymorphism of susceptibility of the strains in the Leishmania world, some differences could be observed between the strains. Therefore, we expect compound 2 to have similar activity in several Leishmania forms and maybe other trypanosomatids, though not on T. brucei (through arginase).

R1.3:  The evaluation of compounds cytoxicity was performed on RAW 264.7 macrophages. It is well known that RAW 264.7 macrophages are immortalized cell lines. Therefore, they could exhibit less sensitivity to some treatments or drug resistance. The use of peritoneal macrophages or bone marrow-derived macrophages could be good and valuable alternatives for further validation assays.

A1.3:  The RAW 264.7 macrophage model is suitable for detecting significative in vitro activity against intramacrophage amastigotes as it is known to be less susceptible to many drugs. Therefore, it is a good model for first screening to detect compounds of interest. We agree that bone marrow-derived macrophages are the most adapted model to decipher further a mechanism of action of the compound.

Reviewer 2 Report

Manuscript Number: molecules-463103

entitled: Synthesis, in silico, and in vitro evaluation of anti-2 leishmanial activity of oxadiazoles and indolizine 3 containing compounds flagged against anti-targets

 I recommend this paper for publishing in Molecules journal after some corrections. I have the following question/comments to the authors.

There is no information about novelty of used chemicals (e.g. acylhydrazones, amidoximes and 1,2,4-oxadiazole) and methodology used. I am afraid they are already reported in literature, so please provide broad readership to these ref.

Authors should compare yours procedure with already reported in literature.

On Scheme 1 should be information about used R1 and R2, and obtained yields. Maybe table should be directly below Scheme 1?

On Scheme 2 and 4 should be information about obtained yields.

Please add some information to the text:

“Melting points (m.p.) were determined using a Mettler Toledo MP50 system” and were uncorrected.

“1H and 13C NMR spectra were recorded in CDCl3 and DMSO-d6 using a Bruker AC 300 (1H) or 75MHz (13C) instruments”. Chemical shifts referenced to ext. TMS (?) (1H, 13C).

Spectral studies must be rewrite.

Present version is very hard to follow. Spectral studies should be more concise, please summarized your data in tables to be more transparent, and will be possible to compare NMR data (chemical shifts and spin systems). Please explain why some chemicals possess 1H NMR and 13C NMR data and some only 1H NMR?

Authors should provide the yield of below’s molecule:

Compound 8 3-[3,4-(Methylenedioxy)phenyl)]-5-(methoxymethyl)-1,2,4-oxadiazole, Compound 11 Dimethyl 3-(5-nitro-1H-benzo[d]imidazol-2-yl)indolizine-1,2-dicarboxylate and Compound 12 Dimethyl 3-(5-nitro-1H-benzo[d]imidazol-2-yl)pyrrolo[2,1-a]isoquinoline-1,2-dicarboxylate, and Compound 13 Methyl 3-(5-nitro-1H-benzo[d]imidazol-2-yl)pyrrolo[2,1-a]isoquinoline-1-carboxylate.

Compound 2 and 4. What does mean „2 br”? and “sL” for 12?

Compound 11. Please check 1H NMR. The multiplets: ddt, ddt and tt are not certain. Please do not use automatic characteristics made by programs (e.g. MestreNowa). Please try to understand the origin of multiplets. The nice program to simulate NMR’s is ACD/LAB. If I am correct please check all NMR’s data.

There is no Conclusion part and Discussion part is too short.

 In my judgment, there is good publishable science in this manuscript, but it needs some work before it can be accepted.

Author Response

Reviewer:

I recommend this paper for publishing in Molecules journal after some corrections. I have the following question/comments to the authors.

R2.1:  There is no information about novelty of used chemicals (e.g. acylhydrazones, amidoximes and 1,2,4-oxadiazole) and methodology used. I am afraid they are already reported in literature, so please provide broad readership to these ref. Authors should compare yours procedure with already reported in literature.

A2.1:  The text has now been modified with more information on the (automated) synthesis, and references included. There is also another part concerning synthesis in the Discussion section.

R2.2:  On Scheme 1 should be information about used R1 and R2, and obtained yields. Maybe table should be directly below Scheme 1?

A2.2:  R1 and R2 are now included in the caption to Scheme 1. Yields obtained are shown in Table 1, which is immediately after Scheme 1.

R2.3:  On Scheme 2 and 4 should be information about obtained yields.

A2.3:  Yields for compounds are now included in the text.

R2.4:  Please add some information to the text: “Melting points (m.p.) were determined using a Mettler Toledo MP50 system” and were uncorrected.

A2.4:  This information has now been added to the text.R2.5:  “1H and 13C NMR spectra were recorded in CDCl3 and DMSO-d6 using a Bruker AC 300 (1H) or 75MHz (13C) instruments”. Chemical shifts referenced to ext. TMS (?) (1H, 13C). Spectral studies must be rewrite. Present version is very hard to follow. Spectral studies should be more concise, please summarized your data in tables to be more transparent, and will be possible to compare NMR data (chemical shifts and spin systems). Please explain why some chemicals possess 1H NMR and 13C NMR data and some only 1H NMR?

A2.5:  We have included in the experimental part the missing 13C of the compounds 1, 2, 4, 6 and 7

The series of each family of compounds presented here is limited; the exposed spectral data are thus all included in the experimental part. Concerning the hydrazone family, the spectra correspond to inseparable mixtures of S-cis and S-trans geometries and sometimes Z/E, as is already known in the literature (Oliveira, P.F.M. et al. Molecules, 2017, 22, 1457). HRMS are correct for all compounds.

R2.6:  Authors should provide the yield of below’s molecule: Compound 8 3-[3,4-(Methylenedioxy)phenyl)]-5-(methoxymethyl)-1,2,4-oxadiazole, Compound 11 Dimethyl 3-(5-nitro-1H-benzo[d]imidazol-2-yl)indolizine-1,2-dicarboxylate and Compound 12 Dimethyl 3-(5-nitro-1H-benzo[d]imidazol-2-yl)pyrrolo[2,1-a]isoquinoline-1,2-dicarboxylate, and Compound 13 Methyl 3-(5-nitro-1H-benzo[d]imidazol-2-yl)pyrrolo[2,1-a]isoquinoline-1-carboxylate.

A2.6: The yields for the compounds have now been included in the text.  

R2.7:  Compound 2 and 4. What does mean „2 br”? and “sL” for 12?

A2.7:  These have been corrected in the text,

R2.8:  Compound 11. Please check 1H NMR. The multiplets: ddt, ddt and tt are not certain. Please do not use automatic characteristics made by programs (e.g. MestreNowa). Please try to understand the origin of multiplets. The nice program to simulate NMR’s is ACD/LAB. If I am correct please check all NMR’s data.

A2.8:  The 1H NMR data have been revised and corrected. The series presented here is quite limited (3 compounds). Concerning the benzimidazole series, some of the authors have synthetized larger family of benzimidazole indolizines and benzimidazole furans and extensive spectral analysis has been made. This part, along with various activities of the compounds should be presented in another manuscript. The focus here being principally on the in silico study and in vitro evaluation. Here again, all HRMS analyses are correct

R2.9:  There is no Conclusion part and Discussion part is too short. In my judgment, there is good publishable science in this manuscript, but it needs some work before it can be accepted.

A2.9:  We thank the Reviewer for their positive assessment of the manuscript and the work within. We have now written a Conclusions section which was optional in Articles in Molecules. 

Round  2

Reviewer 2 Report

Manuscript Number: molecules-463103

Title:  Synthesis, in silico, and in vitro evaluation of anti-2 leishmanial activity of oxadiazoles and indolizine 3 containing compounds flagged against anti-targets

I recommend the present version of manuscript for publishing in Molecules.